# Land Surface Phenology Retrieval through Spectral and Angular Harmonization of Landsat-8, Sentinel-2 and Gaofen-1 Data

**Jun Lu [1], Tao He [1], Dan-Xia Song [2,3,\*] and Cai-Qun Wang [1]**

[1] School of Remote Sensing and Information Engineering, Wuhan University, Wuhan 430072, China; junlurs@whu.edu.cn (J.L.); taohers@whu.edu.cn (T.H.); cqwangrs@whu.edu.cn (C.-Q.W.)
[2] Hubei Provincial Key Laboratory for Geographical Process Analysis and Simulation, Central China Normal University, Wuhan 430079, China
[3] College of Urban and Environmental Sciences, Central China Normal University, Wuhan 430079, China
\* Correspondence: dxsong@ccnu.edu.cn

**Abstract:** Land Surface Phenology is an important characteristic of vegetation, which can be informative of its response to climate change. However, satellite-based identification of vegetation transition dates is hindered by inconsistencies in different observation platforms, including band settings, viewing angles, and scale effects. Therefore, time-series data with high consistency are necessary for monitoring vegetation phenology. This study proposes a data harmonization approach that involves band conversion and bidirectional reflectance distribution function (BRDF) correction to create normalized reflectance from Landsat-8, Sentinel-2A, and Gaofen-1 (GF-1) satellite data, characterized by the same spectral and illumination-viewing angles as the Moderate-Resolution Imaging Spectroradiometer (MODIS) and Nadir BRDF Adjusted Reflectance (NBAR). The harmonized data are then subjected to the spatial and temporal adaptive reflectance fusion model (STARFM) to produce time-series data with high spatio–temporal resolution. Finally, the transition date of typical vegetation was estimated using regular 30 m spatial resolution data. The results show that the data harmonization method proposed in this study assists in improving the consistency of different observations under different viewing angles. The fusion result of STARFM was improved after eliminating differences in the input data, and the accuracy of the remote-sensing-based vegetation transition date was improved by the fused time-series curve with the input of harmonized data. The root mean square error (RMSE) estimation of the vegetation transition date decreased by 9.58 days. We concluded that data harmonization eliminates the viewing-angle effect and is essential for time-series vegetation monitoring through improved data fusion.

**Keywords:** BRDF; spectral and angular harmonization; data fusion; land surface phenology

## 1. Introduction

Land surface phenology (LSP) is an important indicator of climate change [1]. In the past few decades, affected by global climate change, the phenology of terrestrial vegetation has undergone significant changes. European deciduous forest leaf unfolding is 4.2 days earlier every ten years on average [2]. In China, the time of leaf unfolding in deciduous forests is an average of 5.5 days earlier every ten years, which is a greater rate of phenological change than in Europe [3]. Vegetation monitoring can clarify the dynamics of different vegetation types and then reveal the spatial and temporal characteristics of climate change [2]. Therefore, accurate and detailed phenological information is of great value for regional and global climate change studies [4].

Many studies have obtained satellite-based remote-sensing datasets for vegetation monitoring. Data from the Moderate-Resolution Imaging Spectroradiometer (MODIS)

have been widely used to obtain phenological information from different types of vegetation [5,6]. Long-term Data Record (LTDR) Advanced Very High-Resolution Radiometer (AVHRR) observations have been successfully used to monitor distinct phenological events on a global scale [7,8]. The Visible Infrared Imaging Radiometer Suite (VIIRS) instrument onboard the Suomi National Polar-orbiting Partnership (Suomi-NPP) was also used to extend the long-term LSP data records that began with AVHRR and MODIS [9,10]. Although the data have a relatively coarse spatial resolution (≥250 m), the short revisit time gives the data an advantage in LSP retrieval [11,12].

However, owing to the comparatively coarse spatial resolution of these various satellite sensors, detecting phenological dynamics in heterogeneous landscapes can be difficult [10,13]. For a simulation analysis, the variance of the finer pixels in one coarse pixel for the start of the season can be as high as 40 days [14]. The coarse spatial resolution of remote-sensing data makes it difficult to capture the phenology in regions with multiple types of vegetation owing to the mixed-pixel problem, and the ground-measured data cannot be well-matched with the remote-sensing LSP derived by the coarse spatial resolution data [15].

High-resolution data have more detailed spatial information and have been used widely in vegetation phenology mapping [16]. However, high-spatial-resolution data often have a long revisit period, which cannot effectively capture the rapid changes in vegetation [11]. Furthermore, optical remote-sensing systems would face constant data loss in vegetation regions that are often clouded during the growing season. In the mid-latitude region, the average yearly cloud-free probability was 21.3% [17]. The above two points make the use of high-spatial-resolution data for phenological monitoring ineffective.

Integrating imagery from various remote-sensing platforms is a feasible method to compensate for a lack of data, which increases the frequency of observations and therefore collects more data [18,19]. In addition, it uses a variety of data types, including spatial detail [20], temporal information [9], and spectral ranges [21]. Harmonized Landsat Sentinel-2 (HLS) data have been widely used for monitoring vegetation [18,22,23]. Through the combination of the two sensors, a data-set with a repeat cycle of 3.2 days and a 30 m spatial resolution can be formed, which provides a stable data source for high-spatial-resolution LSP monitoring [24]. However, using different types of data in terms of spectral, spatial resolution, and illumination-viewing geometries will likely introduce uncertainties, which will eventually be passed onto vegetation monitoring based on the vegetation index (VI) [10,25]. Therefore, different remote-sensing datasets cannot be combined directly, and the inconsistencies between different datasets should be taken into consideration [26].

Spatio–temporal data fusion is another methodology for integrating imagery from various remote-sensing platforms [27]. Fusion models, such as the spatial and temporal adaptive reflectance fusion model (STARFM), use the spatial and temporal information provided by different satellite data to reconstruct high spatio–temporal resolution data, resulting in higher spatial resolutions and time sampling frequencies for remote-sensing data [28]. However, most applications of STARFM do not consider the difference in the illumination-viewing geometries of the input data [29]. Therefore, the MODIS nadir bidirectional reflectance distribution function adjusted reflectance (NBAR) and Landsat series data provide the major input data for the similar spectral settings and illumination-viewing geometry of these two datasets [28,30]. However, owing to the influence of cloud cover, high-spatial-resolution data are often unavailable. Therefore, the input data sources of the fusion model need to be enhanced to improve the stability of the output results of the model [31,32]. However, the difference between different high-spatial-resolution data in terms of illumination-view geometries and spectral characteristics will increase the uncertainty in the fused result [28].

VI is a useful reference for vegetation monitoring [33], and constructing a time-series VI curve is the main step in phenological retrieval. However, time-series VI is heavily

influenced by the differences in the spectral and illumination-viewing geometries of the observations [34]. Different satellite observations may vary significantly owing to differences in spectral and illumination-viewing geometry. The difference caused by the spectrum is determined by the spectral range and relative spectral response (RSR) [35], and the difference caused by the illumination-view geometry is determined by the surface bidirectional reflectance distribution function (BRDF) [36]. Nagol et al. (2015) [37] revealed that the difference in reflectance before and after the vegetation growing season in the red and near-infrared (NIR) bands of Landsat-5 can exceed 30% caused by the variation in the solar zenith angle (SZA). Moreover, the VI in a nadir viewing usually increases with an increase in the SZA [38]. Yang et al. (2017b) [39] showed that the reflectance of the MODIS red band under the same SZA and relative azimuth angle (RAA) differed by up to 15% with different view zenith angles (VZA).

In this article, we proposed a data harmonization method for normalizing different high-spatial-resolution reflectances with similar spectral and illumination-viewing properties. The STARFM was then applied to the harmonized data to produce time-series and high-spatio–temporal-resolution reflectance data. Finally, the transition date of the typical vegetation types was determined using the fused 30 m spatial resolution reflectance data. The objectives of this study are as follows: (1) to propose a data harmonization algorithm to eliminate the spectral and angular variations among different remote-sensing data, (2) reconstruct the time-series and fine-resolution reflectance data, and (3) investigate the viewing-angle effects in remote-sensing data with different VZAs and improve the accuracy of vegetation phenology detection.

## 2. Materials and Methods

### 2.1. Materials

#### 2.1.1. In Situ Measurement

Near-surface digital repeat photography is an important data source for vegetation monitoring, and can also provide an assessment of phenological information obtained from satellite data [40,41]. The PhenoCam tracks vegetation status every 30 min from dawn to dusk using digital photographs [42] (https://phenocam.sr.unh.edu/webcam/, accessed on 10 February 2022). To ensure the reliability of the conclusions, we monitored different vegetation types across various climatic regions. The in situ measurements were collected from nine PhenoCam sites with different climate and vegetation types (Table 1), and the spatial distribution of in situ sites is shown in Figure 1. Digital photographs were used to provide in situ measurements to evaluate satellite-derived vegetation phenological information. The digital images were acquired between 10:00 a.m. and local solar noon from 1 January to 31 December 2020, every day, which was consistent with the acquisition time of the corresponding remote-sensing data in situ.

**Table 1.** Overview of the nine PhenoCam study sites.

| Site Name | Latitude (°) | Longitude (°) | Elevation (m) | Country | Vegetation Type |
|---|---|---|---|---|---|
| arsbrooks10 (arsb) | 41.9749 | −93.6905 | 312 | USA | agriculture |
| arsmorris2 (arsm) | 45.6270 | −96.1270 | 338 | USA | agriculture |
| burdetterice1 (burd) | 35.8284 | −89.9879 | 70 | USA | agriculture |
| lethbridge (ileth) | 49.7092 | −112.9403 | 950 | Canada | grass |
| millhaft (mill) | 52.8008 | −2.2988 | 137 | UK | deciduous forest |
| montebondonepeat (mont) | 46.0177 | 11.0409 | 1563 | Italy | wetland |
| oakville (oakv) | 47.8993 | −97.3161 | 268 | USA | grass |
| pace (pace) | 37.9229 | −78.2739 | 100 | USA | deciduous forest |
| slovenia2karstsecforest (slov) | 45.5432 | 13.9162 | 436 | Slovenia | deciduous forest |

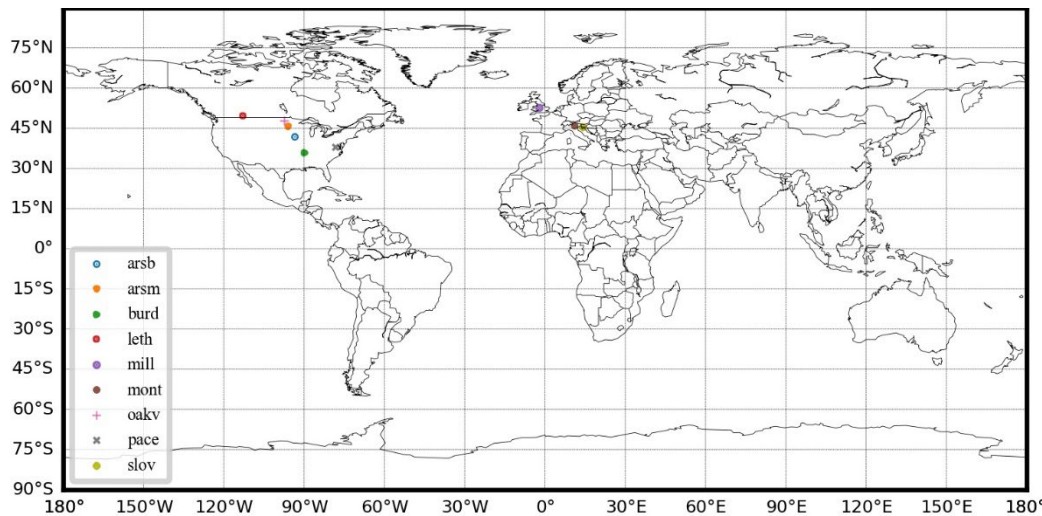

**Figure 1.** The locations of the selected in situ PhenoCam sites.

### 2.1.2. Satellite Data

Multiple types of satellite data were used in this study. The Landsat-8 data used in our study are the atmospherically corrected surface reflectance data downloaded from the United States Geological Survey (USGS) archive (http://earthexplorer.usgs.gov/, accessed on 10 February 2022), generated from the Landsat Ecosystem Disturbance Adaptive Processing System (LEDAPS) [43,44]. The Sentinel-2A Multi-Spectral Instrument (MSI) L2A data were distributed by the European Space Agency Open Access Hub (https://scihub.copernicus.eu/, accessed on 10 February 2022). The Gaofen-1 (GF-1) wide-field viewer (WFV) L1A data were collected from the China Center for Resources Satellite Data and Application (CRESDA) (http://www.cresda.com/CN/, accessed on 10 February 2022), and the surface reflectance of GF-1 was obtained by 6S atmospheric correction model [45]. Detailed information on the above three kinds of data is shown in Table 2, and the red (R) and NIR bands RSR of the different sensors are shown in Figure 2. The auxiliary satellite-based data include the MODIS collection V006 500 m spatial resolution daily gridded NBAR [46] and Ross-Li kernel-driven parameter product (MCD43A1) [36,47], and MODIS aerosol products (MOD04_L2) [48]. NBAR, MCD43A1, and MOD04_L2 were downloaded from NASA's official website (https://search.earthdata.nasa.gov/, accessed on 10 February 2022).

**Table 2.** Basic information about the three fine-resolution satellite sensors.

| | Properties | Landsat-8 OLI | GF-1 WFV | Sentinel-2A MSI |
|---|---|---|---|---|
| Wavelength (nm) | Blue band | 450–515 | 450–520 | 485–523 |
| | Green band | 525–600 | 520–570 | 543–578 |
| | Red band | 630–680 | 630–690 | 650–680 |
| | NIR band | 845–885 | 770–890 | 785–900 |
| Other properties | Spatial resolution (m) | 30 | 16 | 10 |
| | Revisit period (d) | 16 | 2 | 10 |
| | Swath (km) | 185 | 800 | 290 |
| | Quantization (bits) | 12 | 10 | 16 |

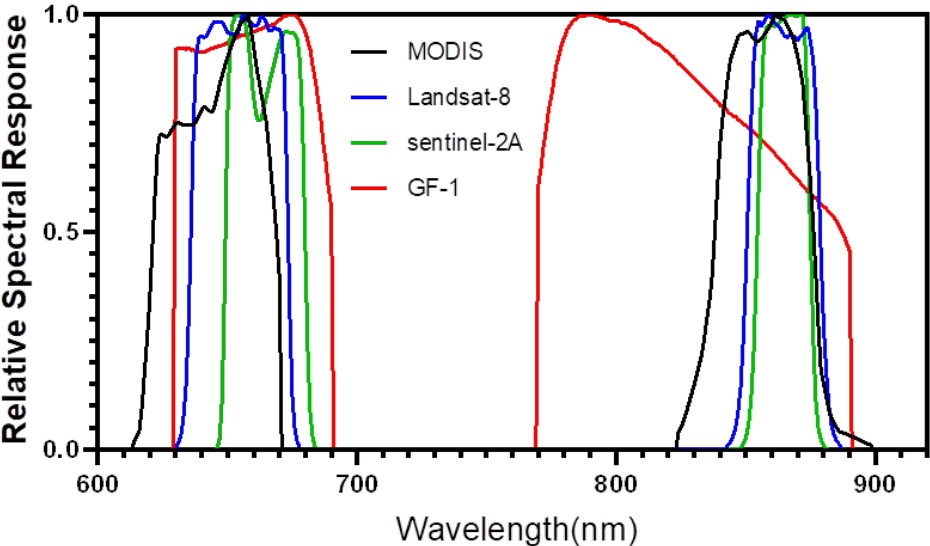

**Figure 2.** Comparison of RSRs for MODIS, OLI, MSI, and WFV.

In this study, the image acquisition time ranged from 1 January to 31 December 2020, and only images without cloud cover in the study area were used. Each PhenoCam site is defined by its center and a 45 × 45 km square (i.e., 1500 × 1500 pixels of Landsat-8 OLI) is taken as the spatial range. The cloudless observations in this range are used as effective observations for vegetation monitoring. The count of effective observations of Landsat-8 OLI, Sentinel-2 MSI, and GF-1 WFV at nine PhenoCam sites is shown in Figure 3a. The day of the year of the three datasets is shown in Figure 3b. It can be seen from Figure 2 that under different climate types, the number of effective observations in each PhenoCam is quite different. The average number of effective observations at the nine stations was 15 in 2020. From the perspective of time distribution, most PhenoCam sites lack effective observations during the start of the season (i.e., March and April). For example, arsb has only one effective observation, while the leth and oakv sites do not have effective observations. Moreover, during the middle of the season (i.e., June and July), some PhenoCam sites lacked effective observations. There were no observations at arsb and mont and only one observation at the burd. The lack of effective observations during the growing season creates uncertainty in remote-sensing vegetation phenology mapping [12]. Therefore, it is critical to reconstruct the intensive observation data.

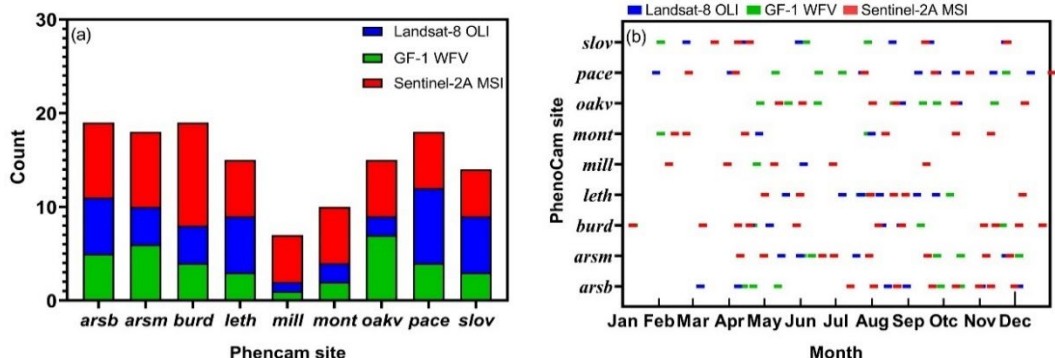

**Figure 3.** Count of effective observations (**a**) and the DOY (**b**) of Landsat-8 OLI, Sentinel-2 MSI, and GF-1 WFV at nine PhenoCam sites. The red is Sentinel-2 MSI, the blue is Landsat-8 OLI, and the green is GF-1 WFV. The name of the site is abbreviated corresponding to the name in Table 1.

*2.2. Methods*

First, the PhenoCam digital photographs were processed to produce time-series ground measurements. Second, a data harmonization method involving band conversion and BRDF correction was proposed to normalize the multisource 30 m satellite data. Daily 30 m reflectance data were further derived by applying the STARFM algorithm to the harmonized data and MODIS NBAR. Third, we extracted the vegetation phenology by applying a piecewise linear fitting algorithm to the daily 30 m VI data and time-series in situ measurements, respectively. Finally, we validated the vegetation transition date derived from the harmonized and unharmonized 30 m data respectively using phenological information extracted from digital repeat photographs. The workflow is shown in Figure 4 and each step is described in the following sections.

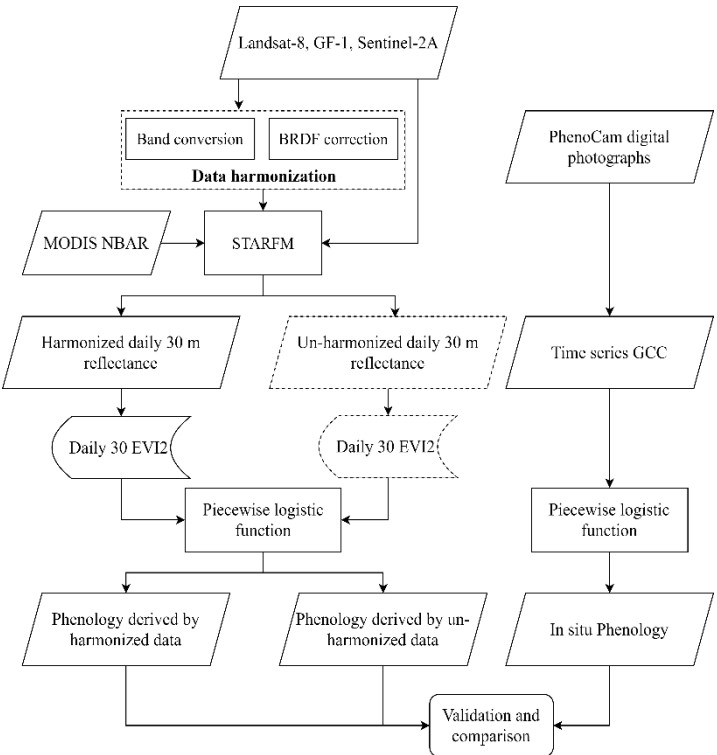

**Figure 4.** Land surface phenology retrieval workflow chart.

2.2.1. Processing In Situ PhenoCam Data

The objective of digital photograph preprocessing is to obtain the green chromatic coordinates (GCC), which is a commonly used index for measuring green vegetation in near-sensing images [42]. The digital number (DN) of the digital photograph was converted to the adjusted DN value using Equation:

$$DN_A = DN/\sqrt{E} \tag{1}$$

where $DN_A$ is the adjusted DN value, $DN$ is the pixel value of the digital photograph, and $E$ is the exposure time, which can be obtained from the metadata of the photo. To eliminate the influence of mixed pixels, we selected the vegetation pixels in the photograph and manually determined the region of interest (ROI).

When the ROI is determined, the GCC of all pixels of the phenological camera photos in the ROI can be calculated using Equation:

$$GCC = \frac{DN_g}{DN_r + DN_g + DN_b} \tag{2}$$

where *GCC* represents the green chromatic coordinates, and $DN_r$, $DN_g$, $DN_b$ are the red, green, and blue components of the adjusted DN value. As the phenology camera made observations every half an hour, and the selected time of the photograph data is 2 h, there will be multiple GCC values for a day. Therefore, the average GCC value for each day is selected as the GCC value of the day.

### 2.2.2. Harmonizing Satellite Data

The inconsistent observations registered by different sensors are mainly due to the different characteristics of the spectral and illumination-viewing geometries [24]. The objective of data harmonization is to solve the problem of inconsistent observations from the input data, which is not addressed in STARFM. Since the Landsat-8 and Sentinel-2A have similar characteristics on overpass time and observation angles [49,50], this study focused on the harmonization of MODIS NBAR and GF-1. The data harmonization includes (1) adjusting data to represent the response from a common spectral band, and (2) normalizing illumination-viewing geometry via BRDF correction. In this study, the MODIS NBAR was chosen as the reference to harmonize the Landsat-8 OLI, Sentinel-2A MSI, and GF-1 WFV. Each substep was elaborated on below.

1. Band conversion

The differences in the observations of different sensors that arise from different RSRs need to be considered. Even for wavelengths designed to have similar coverage, the spectral reflectance can be substantially different because of the RSR [51]. Thus, the sensor signal difference arising from the RSR difference must be considered. As shown in Figure 1, the GF-1 WFV has a broad spectral coverage among the red and NIR bands when compared with the other sensors, and has a significant variation in RSR, especially in the NIR band. Thus, spectral matching should be conducted to mitigate the differences in RSRs. The reflectance for an object in a certain band can be estimated using:

$$\rho = \frac{\int_a^b f(\lambda)\Gamma(\lambda)d\lambda}{\int_a^b \Gamma(\lambda)d\lambda} \tag{3}$$

where is $\rho$ the simulated reflectance of a certain band, $\lambda$ is the wavelength, $\Gamma(\lambda)$ is the relative spectral response, $a$ and $b$ are the spectral range of a specific band, and $f(\lambda)$ is the in situ measured spectrum. In this paper, 245 surface reflectance spectrum samples were collected from the United States Geological Survey (USGS) and advanced spaceborne thermal emission and reflection radiometer (ASTER) spectral libraries [52]. These samples included vegetation, soil, rock, water, snow, and ice [35,53]. The surface spectrum was then used to establish the relationship between the reference sensor and target sensor using a linear regression model:

$$\rho_{\text{MODIS}}(i) = k(j) \cdot \rho(j) + \varepsilon(j) \tag{4}$$

where $\rho_{\text{MODIS}}$ is the simulated reflectance of MODIS, $i$ is the band index of MODIS including red and NIR bands, $k$ and $\varepsilon$ are the band conversion coefficients, and $j$ is the band index corresponding to MODIS red and NIR bands of the high-spatial-resolution data.

2. BRDF correction

The illumination-viewing geometries of the center pixels of each PhenoCam site are plotted in Figure 5. It was observed that the satellite overpass times of all PhenoCam sites were close, and the time difference was approximately 2 h. Among the effective observations of all PhenoCam sites in 2020, the VZAs of Landsat-8 OLI and Sentinel-2A MSI are relatively concentrated, and the values are fixed within 7° and 9°, which can be considered to be nadir viewing sensors. However, the VZAs of GF-1 WFV vary greatly, and the maximum value can exceed 35°. The average difference in the VZAs of the images obtained by the three sensors at all PhenoCam sites was 26.20°. Such a high amplitude of variation in VZAs of the remote-sensing data will inevitably affect the consistency of the

observation results, so the viewing-angle effect between different observations needs to be eliminated.

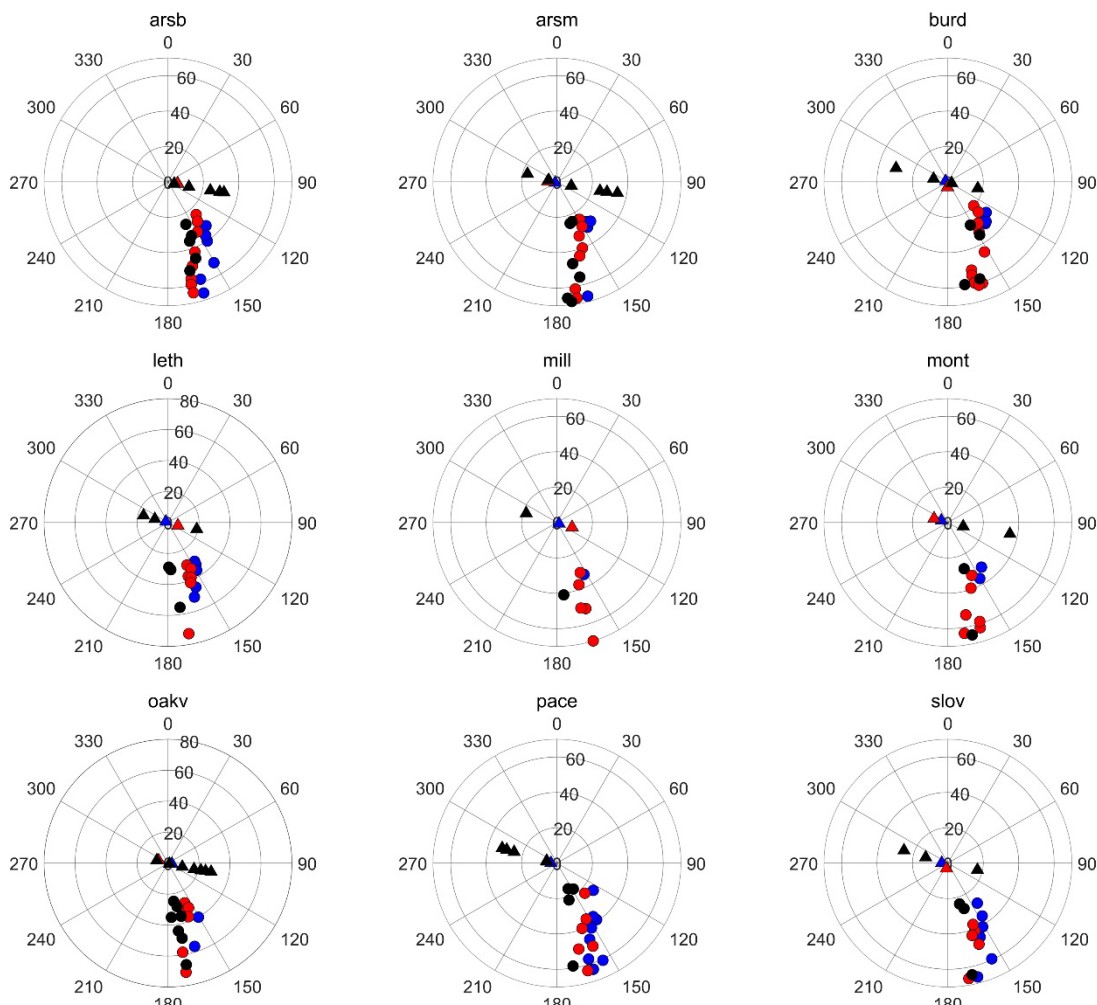

**Figure 5.** Illumination-viewing geometries of different satellite data used in this study. The triangles represent the location of the sensor, and the circles represent the location of the sun. The blue represents Landsat-8 OLI, the red represents Sentinel-2 MSI, and the black represents GF-1 WFV.

To implement the BRDF correction of different data, the following steps were performed: First, MCD43A1 was used to estimate the surface reflectance of the two instruments. The theoretical basis of this product is that the surface reflectance observed from a certain direction can be simulated as a sum of three kernels representing basic scattering types: isotropic, volumetric, and geometric-optical surface [54–56]. The equation is as follows:

$$R(\theta_s,\ \theta_v,\ \varphi,\ \lambda) = f_{iso}(\lambda) + f_{vol}(\lambda)K_{vol}(\theta_s,\ \theta_v) + f_{geo}(\lambda)K_{geo}(\theta_s,\theta_v,\varphi) \qquad (5)$$

where $R$ is the surface directional reflectance; $f_{iso}$, $f_{vol}$, and $f_{geo}$ are the coefficients; $\theta_s$, $\theta_v$, and $\varphi$ are the SZA, VZA, and RAA of the sensor and the sun, respectively. The simulation of surface reflectance in any specific illumination-view geometry can be performed using the MODIS BRDF parameters MCD43A1 ($f_{iso}$, $f_{vol}$, and $f_{geo}$), when the SZA, VZA, and RAA were determined.

We then established the relationship between $R$ and NBAR. The spatial resolution of MCD43A1 is 500 m, and considering the "mixed-pixel" effect, we chose the homogeneous pixels in the R and NBAR to build the BRDF correction model. An objective and automatic

method for selecting homogeneous pixels was proposed in this study. First, a 30 × 30 window was built on the GF-1 WFV surface reflectance based on the selected pixels on *R*. The homogeneous pixels were chosen by the coefficient of variation (CV), which was calculated as follows:

$$CV = \frac{\sigma}{\mu} \tag{6}$$

where $\sigma$ and $\mu$ are the standard deviation and average value of surface reflectance in the window, respectively. If the CV of the pixels within the window is less than 1%, then the coarse MODIS pixel related to the pixels in this window could be considered a homogeneous pixel pair [57].

Due to the non-linear relationship between the directional reflectances under different illumination-viewing geometries of GF-1 WFV and MODIS NBAR, we built a piecewise linear model to calculate the BRDF correction coefficient between *R* and NBAR. It was assumed that the same land cover type has the same BRDF shape, but the magnitude of reflectance may vary [34,58,59]. The normalized difference vegetation index (NDVI) was used as an index to distinguish different land cover types (e.g., bare soils, vegetation, snow/ice, and water). For the same land cover type, variations in the BRDF shape throughout the year were limited and linked to the NDVI [60,61]. The homogeneous pixels were divided into several classes with an interval of 0.2 (>0, 0–0.2, 0.2–0.4, 0.4–0.6, and >0.6) based on NDVI [34]. Then, a linear model was built on each class for each band in *R* and NBAR, as shown in Equation (7). Equation (8) shows a combined set of regression coefficients for all NDVI intervals for a spectral band:

$$R'_{ij} = R_{ij} \cdot P_{ij} + Q_{ij} \tag{7}$$

$$P = P_1 \cup P_2 \cup P_3 \cup \cdots P_i; \ Q = Q_1 \cup Q_2 \cup Q_3 \cup \cdots Q_i \tag{8}$$

where *R′* is the surface reflectance of MODIS NBAR, *P* and *Q* are the BRDF correction coefficients for a specific spectral band and NDVI interval, *i* is the class index, and *j* is the band index. After the linear models of all classes were built, the piecewise linear BRDF correction model for a spectral band in a scene was obtained.

To apply the BRDF correction coefficients derived by the *R* and *R′* to the true observations of GF-1 WFV, we assumed that the BRDF shape is dependent on land cover type and independent of the spatial resolution [62]. Then, we calculated the NDVI of the surface reflectance of GF-1 WFV and divided it into several classes with an interval of 0.2. We applied *P* and *Q* to the same class in each band derived from the surface reflectance of the GF-1 WFV:

$$p_{ij} = P_{ij}; \ q_{ij} = Q_{ij} \tag{9}$$

where *p* and *q* are the BRDF correction coefficients for the surface reflectance of GF-1 WFV. Therefore, the directional reflectance under the illumination-viewing geometry of MODIS NBAR can be simulated by the surface reflectance of GF-1 WFV as follows:

$$\rho(\theta_s, \theta_v = 0, \varphi) = \rho(\theta_s', \theta_v', \varphi') \cdot p + q \tag{10}$$

where $\rho(\theta_s, \theta_v = 0, \varphi)$ is the 16 m spatial resolution harmonized directional reflectance under the illumination-viewing geometry of MODIS NBAR, and $\rho(\theta_s', \theta_v', \varphi')$ is the true surface reflectance of the GF-1 WFV. For a more detailed description of the data harmonization algorithm, we refer to Lu et al. (2022) [26].

### 2.2.3. Generating Time-Series Vegetation Index Data

In this study, the daily 30 m spatial resolution surface reflectance data were reconstructed using STARFM. STARFM is the earliest developed data fusion method based on the weight function which determines how much each neighboring pixel contributes to the estimated reflectance of the central pixel [27]. STARFM assumes that changes in

reflectance are consistent and comparable at different spatial scales over a homogeneous surface. This means that fluctuations existed in coarse spatial resolution data can be introduced directly to the estimation at high resolution.

STARFM uses a linear model to establish the relationship between high-spatial-resolution data (e.g., Landsat) and coarse spatial resolution data (e.g., MODIS). First, MODIS data were reprojected and resampled to fit the Landsat data format. Second, a moving window was built on Landsat data to identify similar neighboring pixels. Third, a weight was assigned to each similar neighbor based on the difference between the surface reflectances of the Landsat-MODIS image pair, which includes the spectral difference, the temporal difference, and the spatial Euclidean distance between the neighbor and the central pixel. Finally, the central pixel can be predicted at a high spatial resolution by the equation below:

$$L\left(x_{\omega/2}, y_{\omega/2}, t_0\right) = \sum_{i=1}^{\omega} \sum_{j=1}^{\omega} \sum_{k=1}^{n} W_{ijk} \times \left(M(x_i, y_{i'}, t_0) + L(x_i, y_{i'}, t_k) - M(x_i, y_{i'}, t_k)\right) \quad (11)$$

where $L$ is the estimated Landsat surface reflectance, $(x_i, y_i)$ is the pixel location, $\omega$ is the window, $t_0$ is the prediction date, $W_{ijk}$ is the weight function, $M$ is the MODIS true observation, and $t_k$ is the true observation date. For a more detailed description of the STARFM algorithm, we refer to Gao et al. (2006) [27].

Several VIs have been implemented to monitor the dynamics of vegetation, including the normalized difference vegetation index (NDVI) [5], enhanced vegetation index (EVI) [63], and soil adjusted vegetation index (SAVI) [64]. The two-band enhanced vegetation index (EVI2) has the capability to reduce background noise and has enhanced sensitivity over dense vegetation canopies. In addition, EVI2 has been shown to have advantages for LSP detection over the commonly used NDVI [65,66]. Therefore, after reconstructing the daily 30 m spatial resolution reflectance data through STARFM, we calculated EVI2 using Equation (12) and constructed the time-series EVI2 data as an indicator of vegetation dynamics.

$$EVI2 = 2.5 \frac{\rho_{NIR} - \rho_{red}}{\rho_{NIR} + 2.4 \cdot \rho_{red} + 1} \quad (12)$$

where $\rho_{NIR}$ and $\rho_{red}$ are the surface reflectance of the NIR and red bands, respectively. The EVI2 time-series usually fluctuates and presents missing values owing to the interference of atmospheric factors such as aerosols, dust, clouds, and snow. Therefore, it was temporally filtered before the phenology extraction. A Savitzky–Golay (S–G) filter is applied first to remove outliers in the EVI2 series. This was performed twice. The local window size was set to 65, 45 and the polynomial degree was set to 2 and 4, respectively. Finally, a gap-filling method based on inverse distance weighted using the two nearest EVI2 values was performed.

### 2.2.4. Vegetation Phenology Detection and Validation

In this study, we estimated the following transition date of the different vegetation types over different regions: (1) greenup, the date photosynthesis started; (2) maturity, the date when the plant has the largest green leaf area, (3) senescence, the date when photosynthetic activity and green leaf area begin to decrease rapidly; and (4) dormancy, the date when physical activity approaches zero [5]. The piecewise logistic function is a widely used method for VI data fitting, and can be modeled using a function of the form:

$$y(t) = \frac{c}{1 + e^{a+bt}} + d \quad (13)$$

where $t$ is time in days, $y(t)$ is the VI data at time $t$, $a$ and $b$ are fitting parameters, $c + d$ is the maximum VI value, and $d$ is the initial background VI value. After curve fitting using the piecewise logistic model, the vegetation transition date can be identified by the curvature and rate of change of curvature. For calculation of the two values, we refer to Zhang

et al. (2003) [5]. For each growth cycle, the rate of change in the curvature of the fitted logistic model was calculated. Four transition dates were then extracted. Greenup and maturity were identified during the growth phase, and senescence and dormancy were identified during the senescence phase.

We calculated the bias and root mean square error (RMSE) to evaluate the vegetation transition date detection results based on multisource remote-sensing data. The vegetation transition dates retrieved by PhenoCam data were taken as the true value, and a comparison of in situ vegetation transition dates with the corresponding pixel in the image was made. The calculation formulas of bias and RMSE are given by Equations (14) and (15):

$$Bias = \frac{1}{n} \sum_{i=1}^{n} \left( \frac{y_i - x_i}{x_i} \right) \tag{14}$$

$$RMSE = \sqrt{\frac{\sum_{i=1}^{n} (y_i - \bar{x_i})^2}{n}} \tag{15}$$

where $x_i$ and $y_i$ are the transition dates of the validation site derived from in situ Pheno-Cam data and remote-sensing data, respectively, $\bar{x_i}$ is the average value of the transition dates, and $n$ is the number of the transition date.

## 3. Results

### 3.1. Data Harmonization Result

The calculation of EVI2 requires information on the reflectance in the red and NIR bands. We used simulated surface reflectance of MODIS as the standard to calculate the conversion coefficients of the red and NIR bands of Landsat-8, Sentinel-2A, and GF-1WFV. As shown in Figure 6, the spectral differences between the red and NIR bands of MODIS and Landsat-8 OLI are relatively small, with the slope of the linear model close to 1, while the spectral differences between the corresponding bands of Sentinel-2A MSI and GF-1 WFV are relatively high.

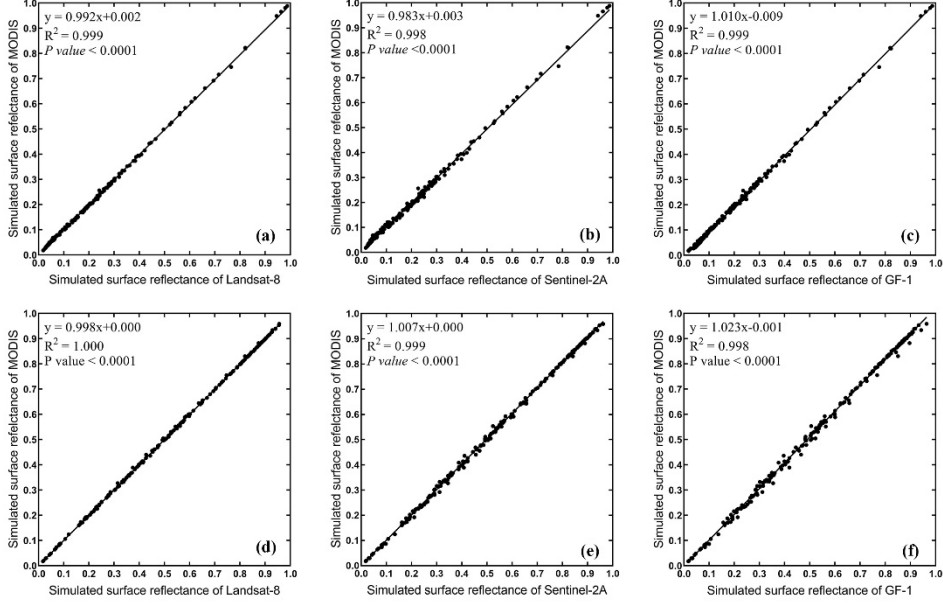

**Figure 6.** Band conversion coefficients of red (**a**)–(**c**) and NIR (**d**)–(**f**) bands of different sensors with the corresponding MODIS bands. The x-axis is the simulated reflectance of the 245 ground types of each sensor, and the y-axis is the simulated reflectance of MODIS.

The input data of STARFM include MODIS NBAR and high-spatial-resolution data, and the VZA of all pixels was 0°. Therefore, in this study, the reflectance of Landsat-8 OLI, Sentinel-2A MSI, and GF-1 WFV was harmonized to the reflectance under the nadir view. We used the Landsat-8 OLI image as the reference to evaluate the data harmonization results of Sentinel-2A MSI and GF-1 WFV as the Landsat-8 OLI is considered a nadir view sensor. To investigate whether data harmonization is beneficial to the accuracy of VI data, we calculated the EVI2 using the Sentinel-2A MSI and GF-1 WFV reflectance with and without BRDF correction and compared it to the EVI2 derived from Landsat-8 OLI. The results are shown in Figure 7.

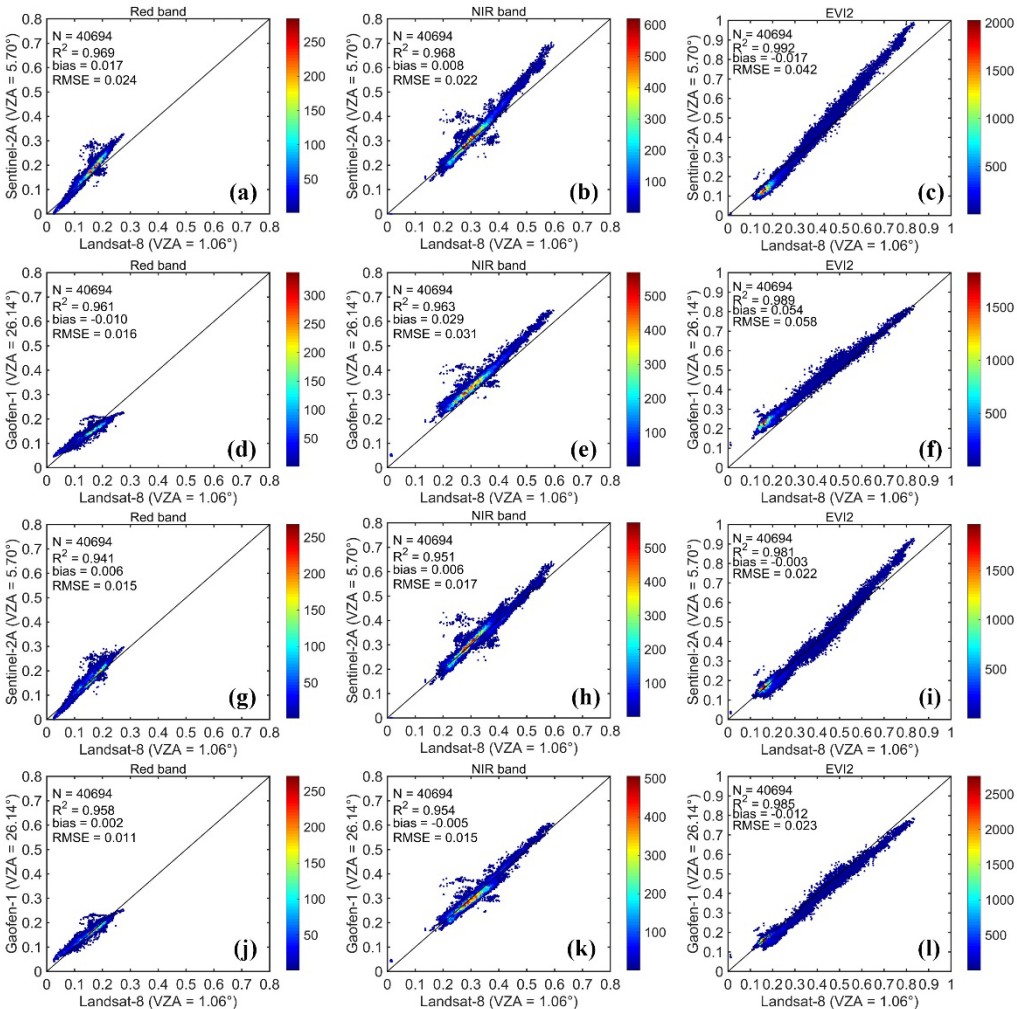

**Figure 7.** Comparison of the surface reflectance and EVI2 without (**a**)–(**f**) and with (**g**)–(**l**) BRDF correction. The GF-1 WFV data (center latitude and longitude: 38.716, −97.021) was obtained on day 323 of 2020. The Landsat-8 OLI data (path/row: 029/034) was obtained on day 325 of 2020, and the Sentinel-2A MSI data (Tile: T14SNG) was obtained on day 323 of 2020.

As shown in Figure 6, the RMSE of the red band reflectance between Sentinel-2A MSI and Landsat-8 OLI decreased by 37.5% while the NIR band decreased by 22.7% after BRDF correction. The RMSE of the red band reflectance between GF-1 WFV and Landsat-8 OLI decreased by 31.3%, while the NIR band decreased by 51.6% after BRDF correction. Moreover, the RMSE of EVI2 between Sentinel-2A MSI and Landsat-8 OLI decreased by 47.6% while the GF-1 WFV dropped by 60.3% after BRDF correction. The improvement of Sentinel-2A MSI was not as obvious as that of GF-1, and the improvement of EVI2 was better than that of reflectance. Therefore, as shown in Figure 6, the data harmonization method proposed in this study can help improve the consistency between the high-spatial-

resolution data and can eliminate the inconsistency in VI caused by the huge difference in illumination-viewing geometries.

To demonstrate the contribution of harmonization to data fusion, we randomly selected a scene of GF-1 WFV data and compared the fused GF-1 reflectance with and without data harmonization to Landsat-8 OLI data. The difference in data acquisition time between GF-1 and Landsat-8 was 4 days, allowing an assumption of no change to the ground. It can be seen from Figure 8 that the $R^2$ of the red and NIR bands are improved by 8.0% and 11.2%, respectively, the bias decreased by 71.1% and 70.8%, and the RMSE decreased by 58.5% and 41.9%, respectively. By inputting the harmonized data into STARFM, the output results on the predicted date could be improved compared with the true reflectance of the predicted date. The improvement of the NIR band was higher than that of the red band. Data harmonization improved the consistency of the fusion result.

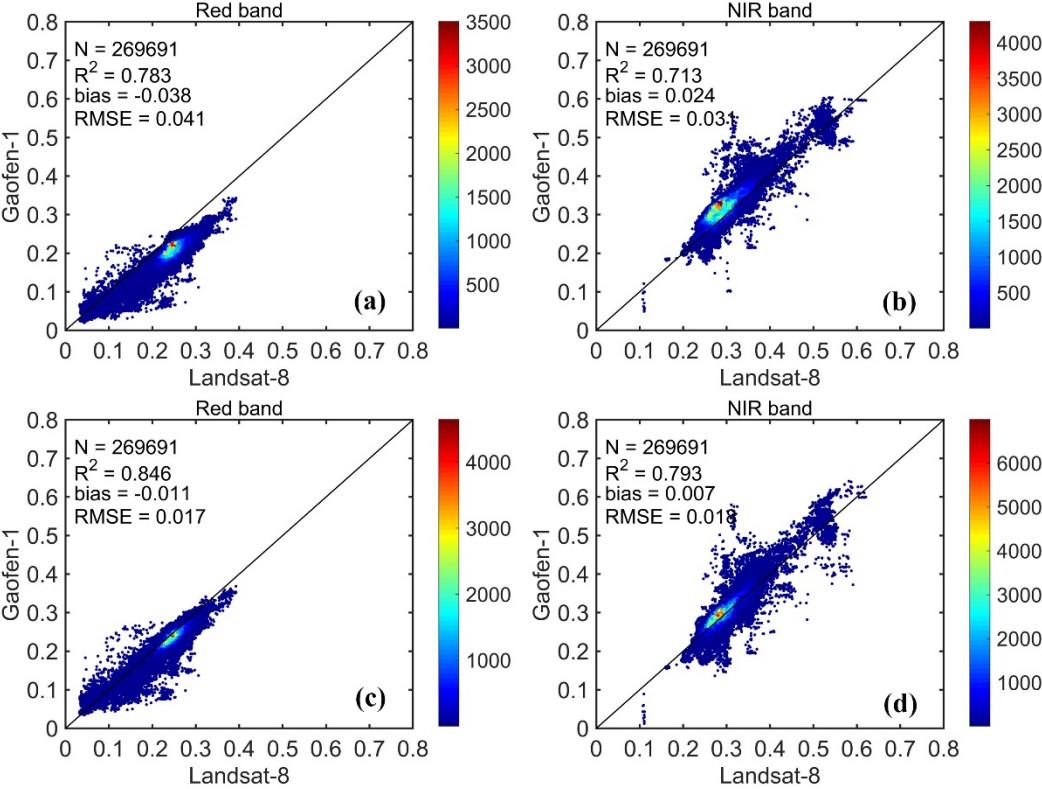

**Figure 8.** Comparison of the fusion result without (**a**,**b**) and with (**c**,**d**) data harmonization. The x-axis ($\rho_{OLI}$) is the reflectance of Landsat-8 OLI on day 237, 2020, and the y-axis is the fused reflectance. The $\rho_{WFV}$ is the fused reflectance with the input of original GF-1 WFV reflectance on day 233, 2020, and the $\overline{\rho}_{WFV}$ is the fused reflectance with the input of harmonized GF-1 reflectance.

To explore whether data harmonization is beneficial for the consistency of the EVI2 curve calculated by the output of STARFM, the high-spatial-resolution data (Landsat-8 OLI, Sentinel-2A MSI, and GF-1 WFV) and the harmonized data were fused with MODIS NBAR to generate two different time-series EVI2 curves. Meanwhile, we calculated the standard time-series EVI2 using the MODIS EVI2 data. As the missing values of the original MODIS NBAR will cause the missing of fusion results, we chose the fusion results of the nine PhenoCam sites to obtain the time-series EVI2 curve in 2020, and the two different types of time-series EVI2 were compared to the MODIS NBAR EVI2 curve. The results are presented in Figure 9. As shown in Figure 9, the $R^2$ of the EVI2 time-series increased by 6.7%, the bias decreased by 18.7%, and the RMSE decreased by 11.3%. Therefore, with the input of the harmonized high-spatial-resolution data, the consistency of the EVI2 curves is improved. Moreover, specific to different vegetation types, the decrease in bias and

RMSE of cropland was 50.0% and 2.5%, respectively, and the improvement in $R^2$ was 3.0%. The decrease in bias and RMSE of grassland was 58.6% and 21.5%, respectively, and the improvement in $R^2$ was 5.0%. As for the deciduous forest, the decrease in bias and RMSE was 123.5% and 10.8%, respectively, and the improvement in $R^2$ was 14.0%. The improvement of the statistical indicators reveals that the consistency of the EVI2 curves was improved with the input of the harmonized data, and this is especially obvious in deciduous forests.

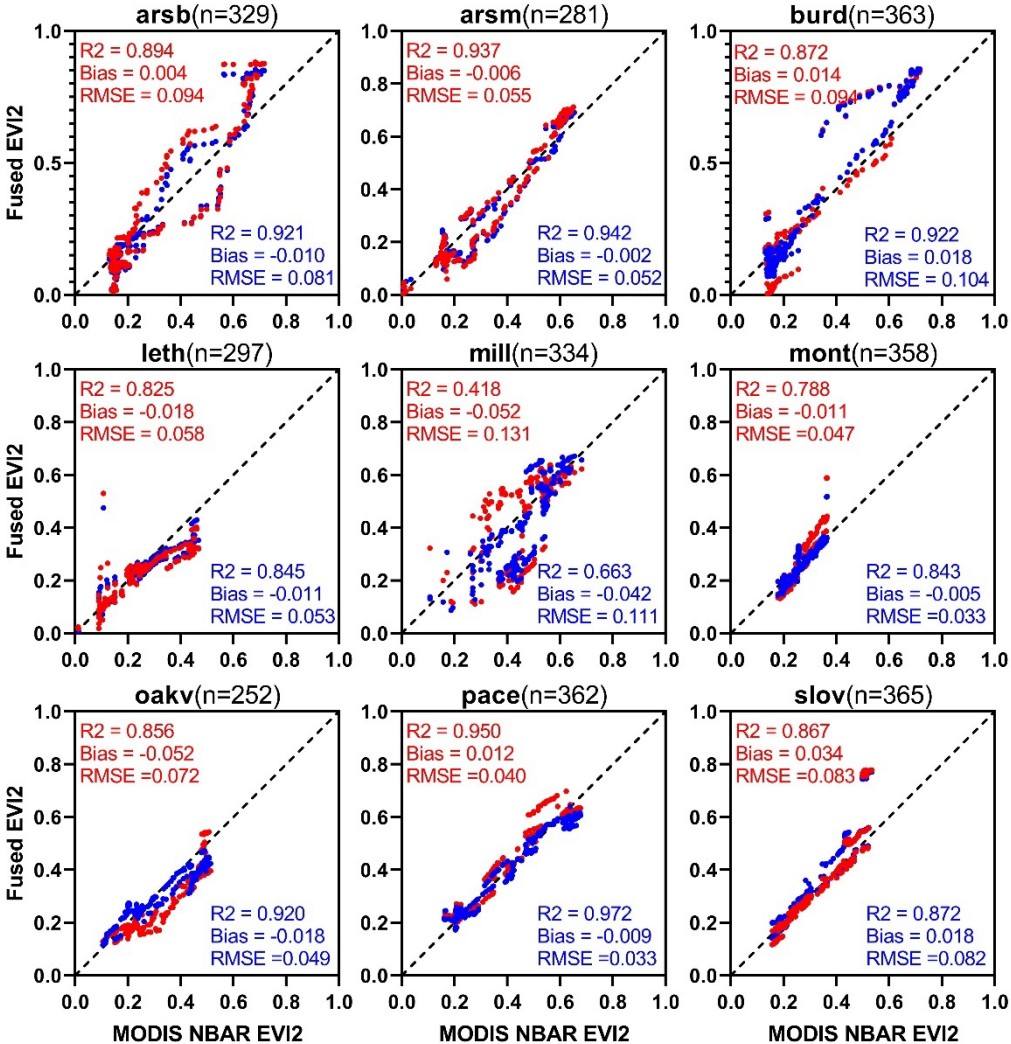

**Figure 9.** Comparison of fused EVI2 in time-series with the input of harmonized and unharmonized data in the nine PhenoCam sites. The red points and the statistical indicators are the fused EVI2 with the input of unharmonized data. The blue points and the statistical indicators are the fused EVI2 with the input of harmonized data. *n* is the count of available fusion results in 2020.

### 3.2. Vegetation Phenology Retrieval Result

The vegetation phenology detection results in this study include in situ and remote-sensing LSP datasets. PhenoCam data are first converted to GCC, and then the transition dates within the year are confirmed by the time-series in situ GCC through the transition date detection method proposed in this study. The in situ phenology detection results are shown in Figure 10.

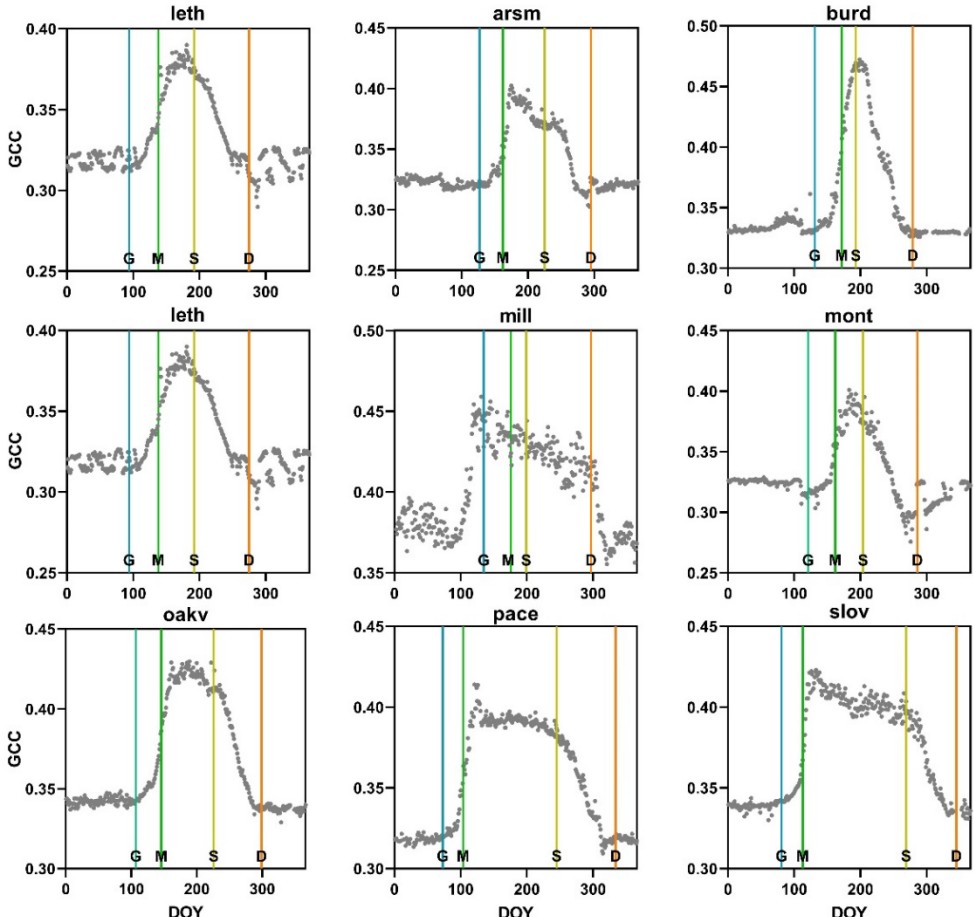

**Figure 10.** Phenological extraction results of the 9 PhenoCam sites. The gray points are daily in situ GCC. The vertical lines are the transition dates. The letters on the vertical lines are G: greenup; M: maturity; S: senescence; D: dormancy.

The single-point phenology extraction algorithm was applied to the daily 30 m spatial resolution fused EVI2 data to achieve high-spatial-resolution phenological mapping in the study area. To investigate the viewing-angle effects on vegetation monitoring using remote-sensing data with large differences in VZAs, we extracted the vegetation phenology using the fused EVI2 constructed by the harmonized and unharmonized data, respectively.

As shown in Figure 11, the filtered EVI2 curve derived from the fused data with and without data harmonization shows a significant difference. The average $R^2$ of the nine PhenoCam sites was 0.94, the average bias was 0.01, and the average RMSE was 0.06. Additionally, the transition dates of the two filtered EVI2 curves also have a large difference, and there is a significant misalignment in the time of peak appearance in the mill and oakv datasets.

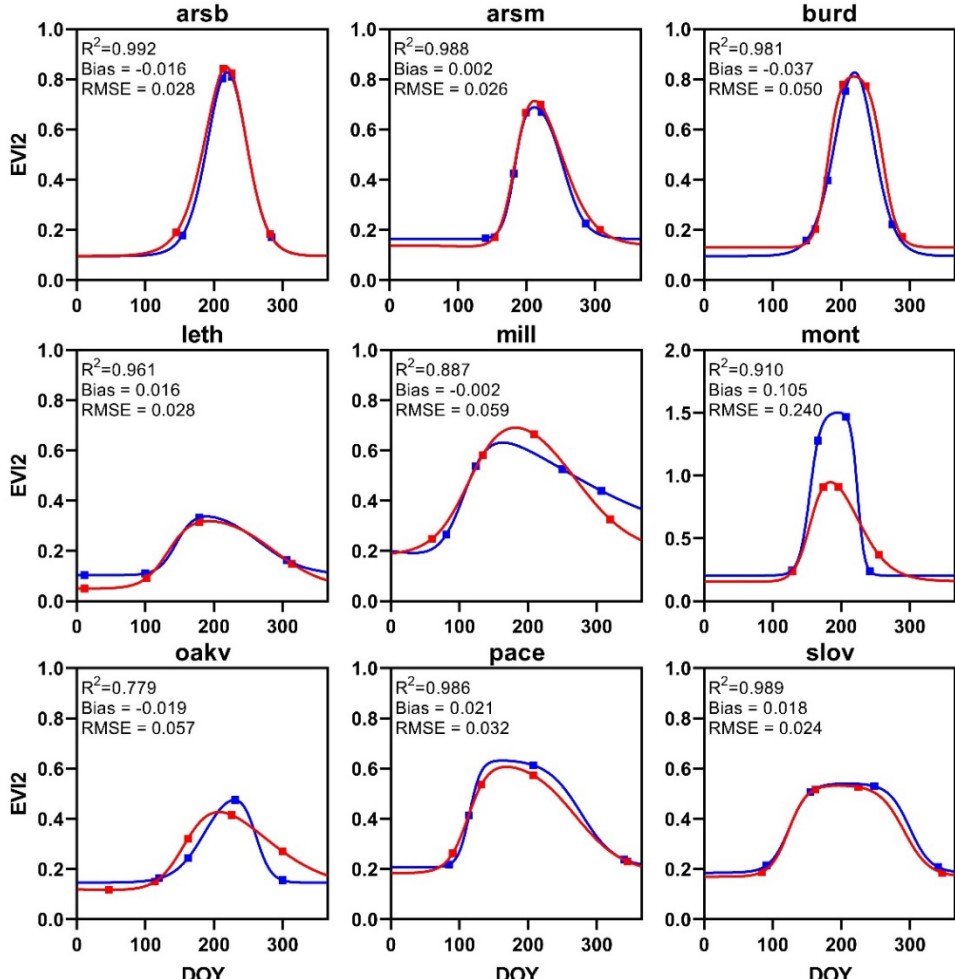

**Figure 11.** Filtered EVI2 curve and the transition date of the nine PhenoCam sites. The red and blue lines are the filtered EVI2 curves derived by the fused data without and with data harmonization, respectively. The red and blue squares are the vegetation transition dates derived by the corresponding EVI2 curves.

In this study, in situ phenology was also used to evaluate the accuracy of phenology data extracted from remote-sensing images. To explore the impact of data normalization on the accuracy of phenology extraction, we compared the two sets of transition dates based on different data in Figure 11 with the results of the in situ measurements. As shown in Figure 12, the $R^2$ improved by 7.5%, the bias decreased by 8.75 days, and the RMSE decreased by 9.58 days. Therefore, with the input of harmonized high-spatial-resolution data, the accuracy of vegetation monitoring was improved.

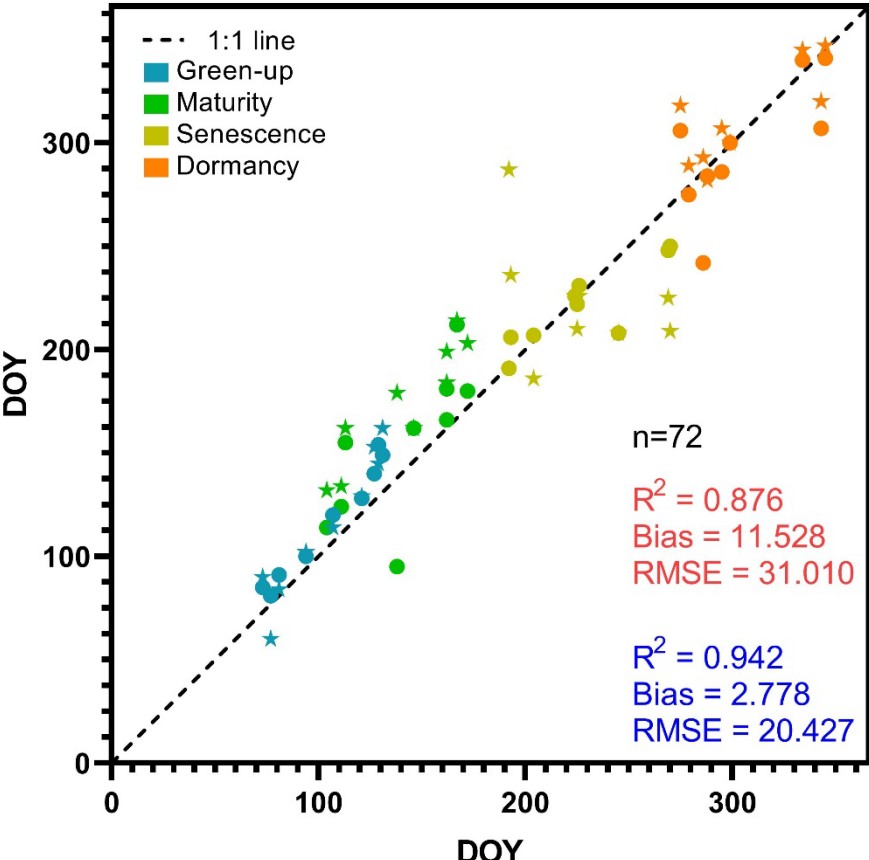

**Figure 12.** Evaluation result based on the in situ transition date. The x-axis is the in situ transition date. The y-axis is the transition date derived by fused data. The stars and circles are the transition date derived by the fused data without and with data harmonization. The red and blue statistical indicators represent the evaluation results of the transition date derived by the fused data without and with data harmonization.

## 4. Discussion

Different illumination-viewing geometries result in inconsistencies in the observation results of the same target introduced by the BRDF effect [34]. When the two viewing angles differ greatly, the BRDF effect in the NIR band is more obvious than that in the red band. In Figure 7, the RMSE of the NIR band of GF-1 WFV and Landsat-8 OLI is higher than that of the red band, and the RMSE of the NIR band of GF-1 WFV is also higher than that of Sentinel-2A MSI. In addition, the difference caused by the BRDF effect is propagated on to the calculation of VI. The RMSEs of EVI2 of Sentinel-2A MSI and GF-1 WFV were higher than the surface reflectance of the red and NIR bands; thus, the difference in EVI2 at different viewing angles even exceeds the band reflectivity (Figure 7). Thus, VI increases the BRDF effect.

Since its establishment, STARFM has been widely implemented [28]. However, when designing the weight function, the variations in the spectral and illumination-viewing geometries between different input data sources were not considered. The input data of STARFM mainly concentrated on the Landsat-8 and MODIS NBAR, owing to the high consistency of the two datasets. In this study, STARFM was implemented on images from two different satellite sensors. These images are different in terms of spatial resolution, bandwidth, spectral response function, atmospheric conditions, and illumination-viewing geometry [24]. Here, we focused on the harmonization of spectral and illumination-viewing geometries between different images. After data harmonization, the consistency between the high-spatial-resolution data and the MODIS NBAR was improved (Figure 6).

The data harmonization method proposed in this paper eliminated the influence of different viewing angles on the VI data. However, the solar angle also affects phenological monitoring [25]. We only harmonized the reflectance of different sensors to a nadir viewing condition and ignored the influence of SZA variation on vegetation monitoring during the year. In the study area, the variation of SZA over the entire year can exceed 60°; therefore, this factor must be taken into consideration for data harmonization.

By comparing the fused time-series EVI2 derived from harmonized and unharmonized data (Figure 9), results showed that in nine PhenoCam sites, the improvement of forest LSP data was greater than that of grassland and cultivated land. This is mainly because the BRDF effect of forest land is more obvious than that of grassland and cultivated land [67]. The data harmonization eliminated the BRDF effect, so the improvement of forest land was more obvious than that of grassland and cultivated land.

In this study, only pure pixels containing a single land cover type were selected to implement and evaluate the BRDF correction. This effectively reduced the impact of the difference in spatial resolution of different sensors. However, the correction effect of pixels was not evaluated in heterogeneous areas. We also assumed that the PhenoCam site and the corresponding pixels were homogeneous, and therefore the transition date retrieval by the PhenoCam site and the corresponding pixels could be compared directly. However, the scale effect of the PhenoCam in situ measurement has not been fully addressed [15]; therefore, an important issue for future research is the spatial representativeness of the PhenoCam in situ measurements as well as the spatio–temporal matching to the remote-sensing data.

## 5. Conclusions

This research proposed a harmonization method to normalize the variations in spectral and illumination-viewing characteristics of high-spatial-resolution reflectance data derived from various satellite sensors. A data fusion method was then performed on the harmonized reflectance data to generate time-series reflectance with high spatio–temporal resolutions. Finally, the daily 30-m reflectance data were used to detect the transition dates of different vegetation types across multiple climate types. It was observed that the viewing angle could affect the accuracy of vegetation phenology monitoring. The proposed data harmonization method can eliminate the inconsistency between fine-resolution reflectance data acquired by different sensors under various illumination-viewing geometries, which further improved the consistency of VI data. The data fusion result of STARFM also benefited from the data harmonization, and the $R^2$ increased by 8.0% and 11.2% for the red and NIR band, respectively. The time-series VI derived from the fused reflectance showed less fluctuations. By using the harmonized reflectance and fused VI data, the accuracy of the vegetation phenology monitoring was improved, with the $R^2$ increased by 7.5% and the RMSE decreased by 9.58 days. We concluded that the proposed data harmonization method should be adopted when using multisource satellite data to monitor vegetation phenology over a large spatial extent.

**Author Contributions:** T.H. and D.-X.S. designed the study; J.L. and C.-Q.W. collected the data, processed the satellite images, and performed the experiments; J.L. conducted the analysis and drafted the manuscript; and T.H., D.-X.S. and C.-Q.W. revised the manuscript. All authors have read and agreed to the published version of the manuscript.

**Funding:** This research was funded by the National Natural Science Foundation of China under Grants 42090012 and 41901300, the Hubei Natural Science Grant 2021CFA082, the Open Fund of Key Laboratory of National Geographical Census and Monitoring, Ministry of Natural Resources (2022NGCM07), and the Fundamental Research Funds for the Central University through Wuhan University under Grant 2042021kf1063.

**Institutional Review Board Statement:** Not applicable.

**Informed Consent Statement:** Not applicable.

**Data Availability Statement:** The PhenoCam digital photographs can be accessed at: https://phe-nocam.sr.unh.edu/webcam/, accessed on 10 February 2022. The Landsat-8 surface reflectance data can be accessed at the United States Geological Survey (USGS): http://landsat.usgs.gov (accessed on 10 February 2022). The Sentinel-2A Multi-Spectral Instrument (MSI) L2A data can be accessed at the European Space Agency Open Access Hub: https://scihub.copernicus.eu/ (accessed on 10 February 2022). The Gaofen-1 (GF-1) wide-field viewer (WFV) L1A data can be accessed at the China Center for Resources Satellite Data and Application (CRESDA): http://www.cresda.com/CN/ (accessed on 10 February 2022).

**Acknowledgments:** The authors would like to thank the Moderate-Resolution Imaging Spectrora-diometer (MODIS) Land Team for providing the aerosol and BRDF products and the three anony-mous reviewers whose constructive comments and suggestions have helped significantly improve the quality of this article. We gratefully acknowledge data support from the National Earth System Science Data Center, National Science & Technology Infrastructure of China: http://www.geo-data.cn (accessed on 10 February 2022).

**Conflicts of Interest:** The authors declare no conflict of interest.

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
