# Peer review of "Land Surface Phenology Retrieval through Spectral and Angular Harmonization of Landsat-8, Sentinel-2 and Gaofen-1 Data"

_remotesensing, doi:10.3390/rs14051296_

Round 1

Reviewer 1 Report

The manuscript with the title “Land Surface Phenology Retrieval through Spectral and Angular Harmonization of Landsat-8, Sentinel-2, and Gaofen-1 data” is an attempt to combine three different satellite sources for vegetation monitoring. The authors used STARFM and MODIS-based data to fuse Landsat-8, Sentinel-2 and Geofen-1 satellite data to a single product and extract vegetation phenology of PhenoCam sites in the Northern Hemisphere. They also validated their results by comparing their harmonized data and derived phenological indices with original data and in situ data.

In general, the paper is presented with a clear and logical structure. The methodology section is well presented with detailed information about the study area and used study methods in this study. The results are good with detailed and well-presented figures. Authors show some discussion to analyse the results and link them with previous studies. The results showed a promising way to harmonize satellite images from three popular sources to better monitor vegetation phenology. While the quality of the paper is generally good, there are still some minor problems/ questions that need to be addressed to avoid confusion before the publication.

  • Why didn’t authors generate the GCC time-series from fused data and derived phenological parameters from that? Could it be better than comparing GCC-based phenological indices with EVI2-based ones?
  • I believe that adding a map with the locations of selected in situ sites would be better for the paper
  • In the figure 5, authors used the term “simulated reflectance of MODIS”. It is unclear for me. Are those reflectances of MODIS NBAR or ones generated by using MCD43A1 datasets and suitable angle parameters?

Reviewer 2 Report

It is in general well written and presents the applied methods and obtained results comprehensively. 

I recommend the manuscript for publication but with minor revisions:

General comments

As a suggestion, I would include materials and methods in one section "Material & Methods". Just to clarify a little more the structure of the paper.

Specific comments

Line 127 / Table 1: Why are these areas chosen? A reason why exactly these areas have been chosen should be included in the text. There is no mention of why this distribution of areas and covers.

Lines 133 – 138: It is not clear if atmospherically corrected reflectances from Landsat 8 are being mixed with L1 products. In the case of all products are without this correction, explain why some type of correction was not used.

Line 158: Only for curiosity: to me, few images seem, what could be due? too much cloudy year? On the other hand, why is only 2020 used as the study period? is it due to needing a high calculation capacity?

Lines 200 – 203: A reference is needed.

Lines 207 – 212: The methodology is not being exposed. It should be moved to the discussion section.

Line 244 / Figure 4: Make the colors stand out more or include fill in the shape. Change the geometry to something more distinguishable than square and circle (triangle or asterisk for example).

Line 324: Has another VI been tried? Or EVI2 has been used directly according to the bibliography.

Line 388 / Figure 6: Modify the figure with a clearer separation between the use or not of BRDF.

Lines 407 – 409: Any reason for choosing this day? Explain please.

Line 412 / Figure 7: Modify the figure with a clearer separation between the use or not of harmonization.

Figures 9 and 10: As a suggestion, perhaps it would be interesting to join both figures. The relationship with the ground truth could be seen better (setting two vertical scales).

Line 493 / Figure 11: Change the geometry to something more distinguishable than square and circle.

Reviewer 3 Report

MS is well written, concise and includes sufficient detail necessary for one skilled in the art to duplicate this work.  The material presented represents an important contribution to the science of landscape phenology prediction using satellite imagery and should be published and it is an important tool for policy makers and other researchers involved in sciences that rely on landscape phenology metrics.

Author Response

MS is well written, concise and includes sufficient detail necessary for one skilled in the art to duplicate this work.  The material presented represents an important contribution to the science of landscape phenology prediction using satellite imagery and should be published and it is an important tool for policy makers and other researchers involved in sciences that rely on landscape phenology metrics.

Response:

Thank you for your review and positive comments on our work. The manuscript has been rigorously checked for English language and spelling, and we believe that the revised version is suitable for publication.